# Efficacy of Early Endoscopic Intervention for Restoring Normal Swallowing Function in Patients with Lateral Medullary Infarction

**DOI:** 10.3390/toxins11030144

**Published:** 2019-03-04

**Authors:** Sun Hyung Kang, Ju Seok Kim, Jong Seok Joo, Hyuk Soo Eun, Eaum Seok Lee, Hee Seok Moon, Seok Hyun Kim, Jae Kyu Sung, Byung Seok Lee, Hyun Yong Jeong, Yeongwook Kim, Min Kyun Sohn, Sungju Jee

**Affiliations:** 1Department of Internal Medicine, School of Medicine, Chungnam National University, Daejeon 35015, Korea; porrtos@daum.net (S.H.K.); showsik@cnuh.co.kr (J.S.K.); jujngs@cnuh.co.kr (J.S.J.); hyuksoo@cnuh.co.kr (H.S.E.); leeusgi@cnuh.co.kr (E.S.L.); mhs1357@cnuh.co.kr (H.S.M.); midoctor@cnu.ac.kr (S.H.K.); jksung69@cnuh.co.kr (J.K.S.); gie001@cnuh.co.kr (B.S.L.); jeonghy@cnu.ac.kr (H.Y.J.); 2Department of Rehabilitation Medicine, School of Medicine, Chungnam National University, Daejeon 35015, Korea; kyu0922@hanmail.net (Y.K.); mksohn@cnu.ac.kr (M.K.S.); 3Daejeon-Chungnam Regional Cardiocerebrovascular Center, Chungnam National University Hospital, Daejeon 35015, Korea

**Keywords:** lateral medullary infarction, botulinum toxin, dysphagia, endoscopy

## Abstract

Dysphagia is considered to be a significant barrier for recovery after lateral medullary infarction (LMI). However, there is still no gold standard treatment for dysphagia. The aim of this study was to explore an effect of an early treatment options for swallowing dysfunction after acute LMI. Medical records of acute LMI patients who had been admitted to the department of rehabilitation medicine from January 2014 to December 2017 were reviewed retrospectively. We compared the clinical efficacy of conventional dysphagia rehabilitation to early endoscopic intervention using either botulinum toxin injection into cricopharyngeal muscle or endoscopic balloon dilatation of the muscle. Outcomes, such as duration of parental feeding, albumin level at diet transition to enteral feeding, and complications, were analyzed. A total of 18 patients with LMI were included. While eight patients (8/9, 88.89%) in the endoscopic group were capable of orally ingesting their diet after intervention, the conversion from tube feeding to an oral diet was possible in only five patients (5/9, 55.56%) of the conventional group during hospitalization. However, the difference between the two groups was not significant (*p*-value ≤ 0.147, chi-square test). Only the final dietary level at the time of discharge was higher level in endoscopic group. The conversion interval from tube feeding to oral diet was also comparable between groups. There was no re-conversion from the oral diet to tube feeding in patients of either group during the median follow-up period of 20 months. Early endoscopic intervention may be a better option for dysphagia with LMI, compared to conventional dysphagia rehabilitation. However, a larger and prospective trial may be needed to confirm our observations.

## 1. Introduction

Lateral medullary infarction (LMI) is a vascular disease of the brainstem. Atherosclerosis of the vertebral artery (VA) or posterior inferior cerebellar artery (PICA) is the most common cause of LMI and is found in 50% of cases. Up to 32 percent of LMI cases are caused by dissection of the VA or PICA, thirteen percent of are caused by small vessel occlusion (SVO), and five percent are caused by cardioembolism and other etiologies [1]. LMI patients with initial dysphagia combined with aspiration pneumonia have been reported to have poor prognosis. Accordingly, these patients need intensive medical care in addition to intensive treatment for dysphagia, such as parenteral feeding, intensive dysphagia rehabilitation, and other nutritional education [2]. LMI infarction affects swallowing function, because the major swallowing centers consisting of the nucleus tractus solitaries (NTS), nucleus ambiguous (NA), and the reticular formation are situated in the dorsolateral medulla oblongata [3]. As a result, dysphagia after LMI may persist or may take months or years to resolve [4].

Dysphagia after LMI appears at the oropharyngeal phase of swallowing [5]. Currently, there are few strategies to treat oropharyngeal dysphagia, including dysphagia rehabilitation therapy, parental feeding, and feeding with a Levine tube. However, these therapies were not able to correct the physiological changes that occur after LMI. Patients often present paralyzed hemipharynx with pharyngo-esophageal dyssynergia. Thus, a recent study evaluated the effects of a surgical pharyngeal resection of the non-contractile pharynx [5]. While the surgical treatment may correct physiological dysfunctions, it is a deterministic and non-refundable treatment for acute stroke patients. Thus, there is still no safe and reversible treatment for dysphagia after LMI. We hypothesized that the closure of the upper esophageal sphincter (UES) after infarction could affect pharyngo-esophageal dyssnergia. Occurrence of UES dysfuction might be the first step, which in turns affect the swallowing process, resulting in dyssynergia. If so, early intervention in the acute or subacute phase of infarction would have a positive impact on rehabilitation and recovery of dysphagia after LMI.

Botulinum toxin (BTX) is a neurotoxin that inhibits tonic and active contraction of muscles by inhibiting release of acetylcholine, and it is useful for managing patients with upper esophageal sphincter (UES) dysfunction [6,7]. BTX was first introduced in 1994 by Schnider et al., and previous studies mostly evaluated treatment with BTX in patients with chronic UES dysfunction [6,7,8,9]. There is no study about early endoscopic intervention for UES in dysphagia after LMI. It is possible that early intervention for dysphagia may influence the clinical course and prognosis of UES dysfunction after LMI. Thus, we performed a retrospective case-controlled study regarding the efficacy of early endoscopic intervention for dysphagia after LMI.

## 2. Results

Eighteen patients with LMI were enrolled in this study. Two patients received EBD treatment alone, three patients received both EBD and botulinum toxin injection, four patients received botulinum toxin injection alone. These nine patients were included in the endoscopic intervention group, and the others were placed in the conventional management group. The conventional group was older than the endoscopic group, and the nutritional state was better in endoscopic group. Initially, none of the patients could sustain an oral diet, and nasogastric tube feeding was provided. Initial video fluoroscopic swallowing study (VFSS), dietary levels, and creatinine levels were not different between groups. Initial albumin and hemoglobin levels were higher in the endoscopic group (Table 1). All endoscopic interventions were done within 30 days after LMI. In the endoscopic group, an oral diet was possible in eight patients (8/9, 88.89%) after endoscopic intervention. Although the conversion from tube feeding to oral diet was possible in five patients (5/9, 55.56%) in conventional group during hospitalization, the difference between the two groups was not statistically significant (*p*-value < 0.147, chi-square test). The final dietary level at the time of discharge was higher in the endoscopic group, and there was difference in the levels of albumin, final hemoglobin level, and follow-up score of VFSS between groups (Table 2). There was no difference in the time interval of conversion to the oral diet between the two groups. There was no re-conversion from the oral diet to tube feeding in patients of either group during the median follow-up interval of 20 months. The initial National Institute of Health Stroke Scale (NIHSS) of the conventional group was higher; however, the final score was not different between groups (Table 1 and Table 2). Initial modified Rankin scale (MRS) score was not different between groups; however, the final MRS score was higher in the conventional group than in the endoscopic group (Table 1 and Table 2).

## 3. Discussion

Swallowing is a complex cooperative process involving the oral, pharyngeal, hypopharyngeal muscles, and the tongue. Up to 70% of patients with stroke might experience dysphagia and oropharyngeal aspiration [10]. The upper esophageal sphincter is comprised of the cricopharyngeal muscles (CPM), inferior pharyngeal constrictor, and the proximal cervical esophagus [11]. CPM is controlled by medulla, thus, medullary infarction may result in a failed relaxation of UES [3]. In several studies, endoscopic management of CPM has shown good clinical results [11,12]. In Alfonsi’s prospective study, botulinum toxin injections were performed in 67 patients, including 14 patients with brain stem infarction. Among 14 brain stem infarction patients, 28.6% showed high response, 50% showed low response, and 24.4% showed no response with injection of botulinum toxin [11]. In a retrospective study of USA, 19 of 24 cricopharyngeal dysfunction patients showed symptomatic improvement after endoscopic balloon dilatation [12]. However, there is no data on the use of early endoscopic intervention after stroke.

There is no consensus about the best time to use endoscopic interventions to treat dysphagia in patients after LMI. Previous studies enrolled patients with more than six months after the appearance of dysphagia symptoms [6,7,8,9,11,12]. We hypothesized that early intervention may have a positive influence on patient’s swallowing function, and that early recovery of swallowing might have positive role on the general condition and rehabilitation outcome of LMI patients with dysphagia. Permanent dysphagia may occur after LMI, and such patients are candidates for intervention targeting CPM. In these patients, conventional swallowing rehabilitation might be insufficient management for recovery of swallowing. Early endoscopic intervention may help patients overcome dysphagia and reduce unnecessary fasting.

Main endoscopic intervention performed in our study was botulinum toxin injection. Botulinum toxin is a family of Clostridial neurotoxin (CNT). CNT has metalloproteases action and blocks neurotransmitter release at nerve terminals. Action of CNT manifest by cleaving several proteins including soluble NSF attachment protein receptor, protein vesicle-associated membrane protein, and synaptosomal-associated protein [13].

In our study, the early intervention group ate a tougher diet than the conventional group (Table 2). Recovery of swallowing function was more frequent in the endoscopic group than in the conventional management group (8/9 versus 5/9); however, there was no statistical difference. Eating is very important factor when assessing quality of life and has a great influence on a patient’s life and mood. We failed to provide evidence showing that early endoscopic intervention significantly improved swallowing; however, the results might be affected by the small sample size and retrospective setting of study. Moreover, we failed to prove that endoscopic intervention was effective for shortening the duration of tube feeding or improving the VFSS score and nutritional statuses of endoscopic intervention patients compared with conventional group. As mentioned above, we assumed that early endoscopic intervention may have positive influence on patient’s swallowing function, and, if so, the patient’s reliance on tube feeding should be reduced. It is possible that the discrepancy between our results and original hypothesis may have originated from our study setting. Further, large scale, prospective studies may prove our hypothesis. Whether to perform conventional rehabilitation or early endoscopic treatment was decided at the discretion of the physician; however, this was decided without clear guidelines, because established guidelines were not available. There was no definite reference to when the most optimal timing to perform the endoscopic intervention, and whether the intervention should be done or not. Treatment decision was mostly made based on physician’s experience and intuition. This may be another weakness of this study.

The age, nutritional status, and NIHSS score of the conventional group showed unfavorable data when compared to the endoscopic group. NIHSS score represents the neurological defects of stroke patients. While such differences may suggest the possibility of selection bias, the MRS score, which represents the degree of disability or dependence in the daily activities, was not different between groups. After the sufficient rehabilitation and medical management, the difference between the NIHSS scores and hemoglobin levels of the two groups were also not significant. MRS score was better in endoscopic group than conventional group after management. However, our results may have been affected by both endoscopic management and possible selection bias.

Our results show that early endoscopic treatment improves the dietary level without any regression in the swallowing function of patients who successfully converted to the oral diet. Thus, early endoscopic intervention may improve swallowing function of LMI patients with dysphagia. While the optimal timing of endoscopic intervention cannot be concluded through our study, we propose that more research using early endoscopic intervention in LMI patients is needed.

## 4. Materials and Methods

### 4.1. Patients and Review Methods

Medical records of LMI patients with dysphagia, admitted at a Tertiary center at Daejeon, Republic of Korea, from January 2013 to December 2017, were reviewed to examine the clinical courses of the patients and collect data. In total, 18 patients were enrolled. Clinical data including VFSS, hemoglobin levels, albumin levels, and diet were examined. Nine patients had undergone endoscopic procedures, such as endoscopic balloon dilatation and botulinum toxin injection, and the other nine patients were managed with conventional swallowing rehabilitation. The primary goal of this retrospective study was to compare the difference in the success rate of switching to an oral diet in the endoscopic intervention group versus the conventional group. Secondary goal was to determine the difference in final dietary levels, time intervals of dietary change, nutritional status, and final VFSS results between the two groups. All patients had LMI within 2 months before start of rehabilitation. There was no difference in time interval from onset of infarction until start of rehabilitation between two groups (early intervention group: 19.5 days (6–29 days) versus the conventional rehabilitation group: 21.2 days (7–41 days), *p* = NS).

### 4.2. Endoscopic Procedure

Endoscopic procedure was performed by a single endoscopist, SH Kang, who has completed more than 5000 gastroscopies. Balloon dilatation was done with a wire-guided balloon dilatation catheters (Boston Scientific, Marlborough, MA, USA), and the balloon size was increased up to 20 mm with six atm of pressure. One hundred units of Botulinum toxin (BOTOX^®^ 100U, Allergan, Dublin, Ireland) was mixed with 2 mL saline and injected under endoscopic guidance (Optimus injector, 23 gauge, Taewoong Incorporation, Seoul, Republic of Korea). Manometry was not performed and UES was found with cap assisted endoscopy (Figure 1). Botulinum toxin was injected separately into four UES quadrants. Initially, EBD was attempted at the earlier period in our institution. However, there were several failed cases of EBD; thus, botulinum toxin injection was tried as a rescue therapy. Many patients showed good response to botulinum toxin injection, thus our policy of endoscopic intervention was changed from EBD to botulinum toxin injection.

### 4.3. Conventional Dysphagia Rehabilitation

Conventional dysphagia rehabilitation included exercise techniques, facilitation techniques, and indirect therapy techniques. Exercise techniques were designed to facilitate the oral motor strength, range of motion, and coordination. Exercise techniques comprised lip exercise, tongue exercise, laryngeal elevation, jaw exercise, respiratory exercise and vocal cord adduction exercise. Facilitation techniques comprised thermal tactile stimulation and inhibitory techniques. Thermal tactile stimulation was tried to stimulate the speed of swallowing. After rubbing the bilateral anterior faucial arch with cold laryngeal mirror or metal rod, swallowing was tried. Inhibitory techniques included inhibition of bite reflex, inhibition of tongue thrust, and facilitation of hypoactive gag reflex. Indirect therapy techniques consist of Shaker exercise and Masako maneuver [14,15].

### 4.4. Video Fluoroscopic Swallowing Study (VFSS)

All patients underwent VFSS that were performed and recorded simultaneously (FLEXAVISION; Shimadzu Corp., Kyoto, Japan). Logemann’s protocol was used to evaluate VFSS [16]. Patients were seated upright on the fluoroscopy chair or wheelchair, and any patient who could not control his or her head independently used a reclined wheelchair. Then, each subject was asked to swallow five types of boluses: Yoplait^®^, soup, 2.5 mL liquid, 5 mL liquid, and rice. To monitor the whole process of swallow, we mixed barium with every food. The amount of barium was about 35%/mL; however, there were some adjustments as we wanted to improve the quality of image. During the entire test period, the sessions were simultaneously recorded as video files at 30 frames per second. The image was viewed in the lateral plane, which included the lips anteriorly to vertebrae posteriorly, and the soft palate superiorly to the sixth cervical vertebra inferiorly. Testing was discontinued if patients showed dyspnea, decrease in saturation, or unstable vital signs. A physiatrist (Sungju Jee) with more than five years of experience performing swallowing studies analyzed the results after VFSS had been finished. All VFSS studies were evaluated by Sungju Jee, according to the American Speech-Language-Hearing Association National Outcome Measurement System Swallowing Scale (ASHA-NOMS). ASHA-NOMS has been widely used for evaluation of patients with dysphagia. It is a rank-score test that measures the degree of dysphagia [17].

### 4.5. Severity of Stoke

Severity of stroke was assessed using the National Institute of Health Stroke Scale (NIHSS) and the MRS. NIHSS is a reliable, scoring system which is used to measure the severity of a stroke [18]. As NIHSS scores rise, the severity of stroke is increased. The MRS score is a six-point disability scale used to measuring the disability or daily activity of patients with stroke [19]. Like NIHSS, a higher MRS score means a more severe disease status than a lower MRS score.

### 4.6. Dietary Level

Liquid diet via tube feeding, dysphagia diet level 1, dysphagia diet level 2, and regular diet was supplied to patients according to their dysphagia level. We applied 0 points to liquid diet, 1 point to dysphagia level 1, 2 points to dysphagia level 2, and 3 points to regular diet. Dysphagia level 1 diet consisted of a puréed diet. The ground food provided to patients was based on a traditional Korean diet. The dysphagia level 2 diet had an increased food viscosity compared to the level 1 diet (Figure 2). A thickener was added to the traditional Korean soup “gook” to prevent aspiration, because it has a higher water content compared to western soups.

### 4.7. Statistical Methods

Statistical analysis was performed using the SPSS software (version 18.0, Chicago, IL, USA, 2009). Chi-square analysis was used to confirm the differences in success rates of the oral diet between the endoscopic intervention group and conventional group. The Mann-Whitney *U* test was used to evaluate the differences between the final dietary level and final VFSS results of the two groups. Differences of age and laboratory tests, including albumin and hemoglobin levels, were evaluated with the Student’s t-test.

### 4.8. Ethical Statement

This study was approved by the Institutional Review Board of Chungnam National University Hospital (IRB number: 2016-12-026) at 01-07-2017, and written consent was waived because of the retrospective design of the study.

## Figures and Tables

**Figure 1 toxins-11-00144-f001:**
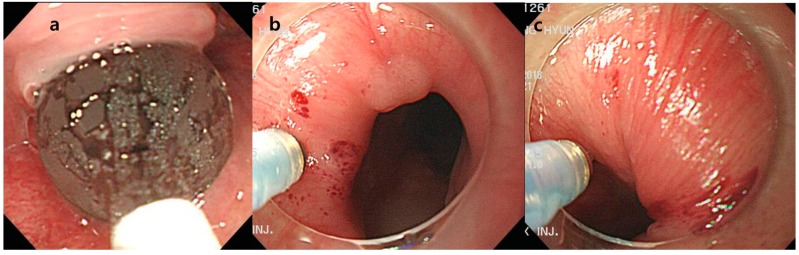
Endoscopic intervention. (**a**) Endoscopic balloon dilatation was performed by dilating a balloon up to 20 mm with 6 atm pressure. (**b**,**c**) Botulinum toxin was injected into four upper esophageal sphincter (UES) quadrants with cap-assisted endoscopy.

**Figure 2 toxins-11-00144-f002:**
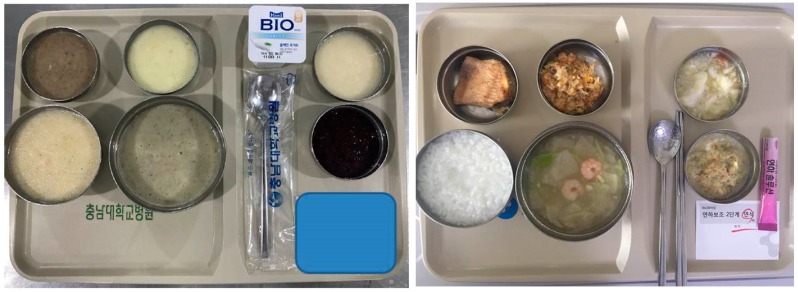
Soft diet for dysphagia patients. (**a**) The dysphagia level 1 diet consisted of a puréed diet. (**b**) The dysphagia level 2 diet had an increased viscousity than that of level 1. Both diets were ground food based on the traditional Korean diet.

**Table 1 toxins-11-00144-t001:** Demographics and Clinical Characteristics.

Characteristics	Endoscopic Group	Conventional Group	*p*-Value
M:F	8:1	3:6	0.01 *
Age (median, minimum to maximum)	56 (46–77)	67 (53–89)	0.015
Hemoglobin (mg/dL, median, minimum to maximum)	13.7 (12.3–17.3)	11.4 (8.7–13.9)	0.010
Albumin (g/dL, median, minimum to maximum)	3.6 (2.9–4.6)	3.3 (3–3.6)	0.022
VFSS (median, minimum to maximum)	1 (1–2)	1 (1–4)	0.224
Dietary level (median, minimum to maximum)	0 (0–0)	0 (0–0)	1.000
NIHSS (median, minimum to maximum)	4 (1–5)	7 (2–19)	0.017
MRS (median, minimum to maximum)	4 (2–5)	5 (3–5)	0.064

* chi-square test. Others were evaluated with Mann Whitney *U* test. VFSS: video fluoroscopic swallowing study; NIHSS: National Institute of Health Stroke Scale; MRS: modified Rankin scale.

**Table 2 toxins-11-00144-t002:** Change of both group after endoscopic intervention and conventional rehabilitation.

Characteristics	Endoscopic Group	Conventional Group	*p*-Value
Hemoglobin (median, minimum to maximum)	13.3 (11.5–17)	11.8 (9.2–14.4)	0.092
Albumin (median, minimum to maximum)	3.9 (3.1–4.4)	3.4 (2.8–3.7)	0.005
VFSS (median, minimum to maximum)	5 (3–6)	5 (1–6)	0.519
Success of conversion from tube feeding to oral diet	8/9	5/9	0.147 *
Dietary level (median, minimum to maximum)	2 (0–3)	1 (0–2)	0.017
Time interval from tube feeding to oral diet (days, median, minimum to maximum)	16 (1–28)	23 ((12–27)	0.826
NIHSS (median, minimum to maximum)	1.5 (1–4)	6 (0–10)	0.209
MRS (median, minimum to maximum)	1.5 (1–4)	4 (2–5)	0.026

* chi-square test. Others were evaluated with Mann Whitney *U* test.

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
