# Peer review of "Efficacy of Early Endoscopic Intervention for Restoring Normal Swallowing Function in Patients with Lateral Medullary Infarction"

_toxins, 2019, doi:10.3390/toxins11030144_

Round 1
Reviewer 1 Report
This manuscript discussed about the assessment of efficacy of two different interventions to treat dysphagia caused by lateral medullary infarction (LMI). These two interventions are conventional and early endoscopic intervention respectively. In these two different treatments, various parameters were evaluated, including success of conversion from tube feeding to oral diet and time interval from tube feeding to oral diet, which are the most related indexes for recovery and showed no significant difference between two groups. This is kind of disappointing. Nevertheless, the rationale to treat dysphagia via early endoscopic intervention is not convincing and more detail information has to be addressed in the introduction. This reviewer also has following concerns:
1. Line 10-11. This sentence is confusing. Please rewrite to clarify specifically what are strategies for conventional dysphagia rehabilitation and early endoscopic intervention.
2. Line 42. Consider to remove “the target of”.
3. Line 63. Please give the full name of VFSS. Same for NIHSS in line 73 and MRS in line 74.
4. Line 87. Please explain in detail that in what disease treatment endoscopic management of CPM has shown good clinical results and what have been improved after treatment.
5. Line 88. This author claimed early intervention and this lead reviewer to assume endoscopic interventions have been done at late stage for patients after stroke in reported studies. Can author give some detail information about the time point for treatment happened in this study and reported studies? Please also define in what time point intervention which has been done belongs to early intervention.
6. Line 90 please add “than” after “more”.
7. Author has to supply more information about conventional intervention in the introduction part and tell more about how the conventional interventions have been done in this study. Why some patients got EBD but others are treated with botulinum neurotoxin? What is the rationale to do this?
Author Response
Response to reviewer 1
Thank you for the valuable review and insightful comments. I replied to each comment and question thoroughly. I revised my manuscript based on your comments and all corrections made in the revised manuscript were highlighted with red color.
This manuscript discussed about the assessment of efficacy of two different interventions to treat dysphagia caused by lateral medullary infarction (LMI). These two interventions are conventional and early endoscopic intervention respectively. In these two different treatments, various parameters were evaluated, including success of conversion from tube feeding to oral diet and time interval from tube feeding to oral diet, which are the most related indexes for recovery and showed no significant difference between two groups. This is kind of disappointing. Nevertheless, the rationale to treat dysphagia via early endoscopic intervention is not convincing and more detail information has to be addressed in the introduction. This reviewer also has following concerns:
I agree with your opinion. To further support and strengthen my article, I added more details about our assumption in the Introduction. (We hypothesized that the closure of the UES after infarction could affect pharyngo-esophageal dyssynergia. Occurrence of UES dysfunction might be the first step, which in turns affect the swallowing process, resulting in dyssynergia. If so, early intervention in the acute or subacute phase of infarction would have a positive impact on rehabilitation and recovery of dysphagia after LMI.)
1. Line 10-11. This sentence is confusing. Please rewrite to clarify specifically what are strategies for conventional dysphagia rehabilitation and early endoscopic intervention.
à I corrected that sentence in the abstract according to your comments.
We compared the clinical efficacy of conventional dysphagia rehabilitation and early endoscopic intervention using either botulinum toxin injection into the cricopharyngeal muscle or endoscopic balloon dilatation (EBD) of the muscle.
2. Line 42. Consider to remove “the target of”.
à I removed “the target of” as you mentioned.
3. Line 63. Please give the full name of VFSS. Same for NIHSS in line 73 and MRS in line 74.
à Full names of VFSS, NIHSS, and MRS were added in the article.
4. Line 87. Please explain in detail that in what disease treatment endoscopic management of CPM has shown good clinical results and what have been improved after treatment.
à I added detailed information about clinical results of these studies in the Discussion part. In Alfonsi’s prospective study, botulinum toxin injections were performed in 67 patients, including 14 patients with brain stem infarction. Among 14 brain stem infarction patients, 28.6% showed high response, 50% showed low response, and 21.4% showed no response with injection of botulinum toxin.[11] In a retrospective study of U.S., 19 of 24 cricopharyngeal dysfunction patients showed symptomatic improvement after endoscopic balloon dilatation.[12]
5. Line 88. This author claimed early intervention and this lead reviewer to assume endoscopic interventions have been done at late stage for patients after stroke in reported studies. Can author give some detail information about the time point for treatment happened in this study and reported studies? Please also define in what time point intervention which has been done belongs to early intervention.
à I added more detailed information about time point for treatment in the Materials part. Patients: All patients had LMI within 2 months before start of rehabilitation. There was no difference in time interval from onset of infarction until the start of rehabilitation between two groups (early intervention group: 19.5 days (6-29 days) vs conventional rehabilitation group: 21.2 days (7-41 days), p=NS).
6. Line 90 please add “than” after “more”.
à I added “than” after “more”.
7. Author has to supply more information about conventional intervention in the introduction part and tell more about how the conventional interventions have been done in this study. Why some patients got EBD but others are treated with botulinum neurotoxin? What is the rationale to do this?
I added more detailed explanations about the conventional dysphagia rehabilitation in the Method section. 4.3 Conventional dysphagia rehabilitation
Conventional dysphagia rehabilitation included exercise techniques, facilitation techniques and indirect therapy techniques. Exercise techniques were designed to facilitate the oral motor strength, range of motion and coordination. Exercise techniques comprised lip exercise, tongue exercise, laryngeal elevation, jaw exercise, respiratory exercise and vocal cord adduction exercise. Facilitation techniques comprised thermal tactile stimulation and inhibitory techniques. Thermal tactile stimulation was tried to stimulate the speed of swallowing. After rubbing the bilateral anterior faucial arch with cold laryngeal mirror or metal rod, swallowing was tried. Inhibitory techniques included inhibition of bite reflex, inhibition of tongue thrust, and facilitation of hypoactive gag reflex. Indirect therapy techniques comprised Shaker exercise and Masako maneuver.[14,15]
è Unfortunately, since our study was done retrospectively, it was not always possible to figure out why some patients were treated with EBD while others were treated with botulinum toxin injection. Physicians relied more on personal experience when making the treatment decision due to the lack of established guidelines. Most treatment decisions seemed to be selected arbitrarily. I added more detailed information about this in the Endoscopic procedure section.
è Endoscopic procedure (Method): Initially, EBD was attempted at the earlier period in our institution. But there were several failed cases of EBD, so botulinum toxin injection was tried as a rescue therapy. Many patients showed good response to botulinum toxin injection, so our policy of endoscopic intervention was changed from EBD to botulinum toxin injection.
Reviewer 2 Report
The authors compared early treatment option for swallowing dysfunction after acture lateral medullary infraction (LMI). The study is well-performed and the material and methods section well-organized. Morreover, in the discussion section, the readers will find all the limitations of the current study. However, prior publication, I will suggest to include more information regarding botulinum neurotoxins (mechanism of action, classification and pharmacology). Please refer to: Azarnia Tehran et al., Toxins 2018 and Pirazzini et al., Toxicon 2018.
Author Response
Response to reviewer 2
Thank you for your thoughtful review and informative comments. I replied to each comment and question thoroughly. I revised my manuscript based on your comments and all corrections made in the revised manuscript were highlighted with red color.
The authors compared early treatment option for swallowing dysfunction after acute lateral medullary infraction (LMI). The study is well-performed and the material and methods section well-organized. Moreover, in the discussion section, the readers will find all the limitations of the current study. However, prior publication, I will suggest to include more information regarding botulinum neurotoxins (mechanism of action, classification and pharmacology). Please refer to: Azarnia Tehran et al., Toxins 2018 and Pirazzini et al., Toxicon 2018.
-> I added more information regarding the mechanism of action and classification of botulinum toxin in the Discussion part and added the reference: Pirazzini M, et al. Hsp 90 and thioredoxin-thioredoxin reductase enagle the catalytic activity of clostridial neurotoxins inside nerve terminals. Toxicon, 2018;147:32-7.
Round 2
Reviewer 1 Report
Manuscript is much improved. This reviewer has no further comments.